# What News Do People Get on Social Media? Analyzing Exposure and Consumption of News through Data Donations

## ABSTRACT

Understanding how exposure to news on social media impacts public discourse and exacerbates political polarization is a significant endeavor in both computer and social sciences. Unfortunately, progress in this area is hampered by limited access to data due to the closed nature of social media platforms. Consequently, prior studies have been constrained to considering only fragments of users' news exposure and reactions. To overcome this obstacle, we present an innovative measurement approach centered on donating personal data for scientific purposes, facilitated through a privacy-preserving tool that captures users' interactions with news on Facebook. This approach offers a nuanced perspective on users' news exposure and consumption, encompassing different types of news exposure: selective, incidental, algorithmic, and targeted, driven by the diverse underlying mechanisms governing news appearance on users' feeds. Our analysis of data from 472 participants based in the U.S. reveals several interesting findings. For instance, users are more prone to encountering misinformation because of their active selection of low-quality news sources rather than being exposed solely due to friends or platform algorithms. Furthermore, our study uncovers that users are open to engaging with news sources with opposite political ideology as long as these interactions are not visible to their immediate social circles. Overall, our study showcases the viability of data donation as a means to provide clarity to longstanding questions in this field, offering new perspectives on the intricate dynamics of social media news consumption and its effects.

## 1 INTRODUCTION

Over the past two decades, there has been a sea change in how people consume news. Earlier, people had to search and actively *select* the news sources they would like to read from, leading to *selective exposure* [19]. More recently, with the growing popularity of social media, news has started to appear on people's social media feeds as a byproduct of their social relations (i.e., posts shared by friends) and recommendation algorithms (i.e., posts recommended by the platforms). In this new media environment, people are unintentionally exposed to news during their diverse online interactions [29, 56, 58].

A vast literature in both social and computer science has since studied the impact of such *incidental exposure* on public discussion quality (abundance of fast, junk, or fake news) [20, 21, 54], and to which extent it may exacerbate political polarization and filter bubbles [30, 36]. The findings are mixed with evidence suggesting

Permission to make digital or hard copies of part or all of this work for personal or classroom use is granted without fee provided that copies are not made or distributed for profit or commercial advantage and that copies bear this notice and the full citation on the first page. Copyrights for third-party components of this work must be honored. For all other uses, contact the owner/author(s).
*Conference'17, July 2017, Washington, DC, USA*
© 2023 Copyright held by the owner/author(s).
ACM ISBN 978-x-xxxx-xxxx-x/YY/MM.
https://doi.org/10.1145/nnnnnnn.nnnnnnn

both (a) incidental exposure leads to receiving information from a significantly narrower spectrum of sources compared to web search [35], and (b) social media users get exposed to significantly more news sources than people who do not use social media at all [45]. While the topic has received wide attention, research has been hindered by the *lack of access to data due to the closed nature of online platforms* – external researchers do not have access to what news users see on social media. We tackle this limitation in this work by proposing a measurement methodology that can *provide a comprehensive picture of users' news exposure on social media and capture user interactions at a finer granularity*.

The *first contribution* of our paper is to present a measurement methodology based on *donations of personal data for scientific research* that allows us to study exposure and consumption of news on social media in a realistic and fine-grained manner. The key to this methodology is to enable users to donate their data in an automated and inconspicuous manner that does not disturb their regular activity. We enable this by building a monitoring tool CheckMyNews which is able to capture, in the background, all posts related to news that appear in users' Facebook feeds, how users interact with them (e.g., share, click), and the news articles users read on news websites outside of Facebook. CheckMyNews ensures that data donations are pseudonymized and do not include any non-news private posts or data from friends. We posted about the tool and the surrounding concept of data donation on Prolific; 889 U.S. residents agreed to install our tool and keep it active for six weeks. For the analysis, we filtered out users with minimal activity on Facebook, ending up with 472 users exposed to 143,129 *news posts* during the data collection period (November 2020 to February 2021).

To capture a wide range of news posts, we aggregated a list of over 12,000 U.S. news sources for our tool to monitor. We first consult two independent news ecosystem auditors – Media Bias Fact Check [32] and News Guard [34] – who list a total of 4,149 news domains. We further develop a method to discover an additional 8,084 *under-the-radar* news sources not listed by journalistic authorities but that claim to be news organizations on Facebook. *To our knowledge, this is the most extensive compilation of U.S. outlets claiming to be news organizations.*

The *second contribution* of the paper is to use the *realistic* and *fine-grained* representation of users' news exposure and consumption to answer several longstanding questions about social media news that have been only partially answered till now, due to the lack of access to data. First, we measure the *political diversity* of users' news diets on Facebook – how much they are exposed to a varied political spectrum. Second, we assess the *quality* of news users receive and how prevalent misinformation is on their feeds. Finally, we measure how users engage with news and the extent to which *exposure to news transforms into real consumption*.

Contrasting (some) previous works [8, 45], we argue in this paper that we need to move beyond treating the news exposure of

users on social media as a whole, as there are different *underlying mechanisms* through which news appears on users' Facebook feeds. First, users can choose to follow news media outlets on Facebook, and as a result, they will see news posts from these outlets in their feeds. This constitutes a form of *selective news exposure.* Second, users can see news posts in their feeds because their friends or communities share them. This is classically considered as *incidental news exposure.* Third, some news posts appear in users' feeds because the platform predicts that the users might be interested in such posts based on their past behavior. These posts are labeled as "Suggested for you" posts on Facebook. We call this *algorithmic news exposure.* And finally, much less realized by the community, there is a fourth type – the *targeted news exposure.* Targeted news exposure is brought by the emergence of online advertising where *self-interested third parties can pay ad platforms to show specific news to particular groups of people.*

To understand the impact of these different types of exposure, we assess the diversity, quality, and consumption of news per underlying mechanisms and make the following observations:

• On average, 5.1% of news posts users encounter on Facebook are from sources known to post misinformation repeatedly. When examining each category separately, we find that selective news exposure has the highest fraction (5.8%) of posts from misinformation sources, while targeted news exposure has the lowest fraction (2.5%). These results suggest that users are more likely to expose themselves to sources known to spread misinformation than be exposed to them due to their friends or the platform's algorithms.

• Targeted, algorithmic, and incidental news exposures are significantly more politically balanced than selective exposure, indicating that while users actively subscribe to news sources of the same political leaning, they get exposed to sources from the opposite political leaning through other mechanisms on Facebook.

We further analyze three types of interactions with news posts: (i) *visible interactions* like commenting, sharing, or liking a post that are visible to other Facebook users, (ii) *hidden interactions* that are not visible to a user's friends, such as clicking on the post to visit the actual article, clicking on the Facebook page of the publisher, or saving the post, and (iii) *visibility time* that captures the time a post was visible on the screen of the user.

• We find that users interact with only 5.1% of news posts they see on Facebook (visible interactions on 2.6% and hidden interactions on 2.8% of news posts). Users had both visible and hidden interactions on < 0.5% of posts. Interestingly, users accessed the landing URLs of only 14% of the news posts they shared. This indicates that *users mostly limit themselves to reading the news post's text and/or images before sharing rather than reading the actual news article.*

• While the fraction of visible interactions on selective and incidental news posts is close to the fraction of hidden interactions, the fraction of visible interactions on algorithmic and targeted news posts is 1.8 to 6 times lower than the fraction of hidden interactions. This suggests that algorithmic and targeted exposure might have an inhibitory effect on users' willingness to share or comment on news posts publicly.

• Finally, we observed more *visible* interactions on posts published by sources with a matching political ideology than the opposing ideology of a user. For instance, Republicans interacted with 3.1% of

posts from the right-leaning media vs. 1.8% from left-leaning ones; whereas Democrats interacted with 3.9% of posts from left-leaning sources vs. 3.1% from right-leaning ones. Surprisingly, when it comes to *hidden* interactions, we observed that both Republicans and Democrats interacted more with posts with opposing political ideology. For example, Republicans interacted with 4.1% of posts from left-leaning sources vs. 2.3% from right-leaning media, while Democrats interacted with 2.9% of posts from right-leaning sources vs. 2.4% from left-leaning ones. This suggests that *users are indeed willing to engage with opposing views, albeit in a private manner.*

Besides answering multiple open questions in the literature related to how users interact with news posts, our work shows that data donations from social media users are both feasible and critical in uncovering the impact of current technologies on society, as well as for the advancement of scientific research. Our codebase is publicly available[1] to be audited and to encourage other researchers to build on this methodology.

**Related Work.** Prior works have attempted to reconstruct users' news exposure using *computational* [3, 4] and *survey-based methods* [6, 19, 37]; but both approaches provide only an *incomplete* picture. For example, one computational approach suggests using a user's public activities (e.g., all news articles they publicly shared or commented on Twitter or Facebook) to reconstruct their news exposure [3, 9, 21]. This approach introduces two kinds of biases. First, it can only consider users who publicly share content, while prior studies have shown that only a small fraction of users take such explicit actions [57]. Second, it can only observe the content users are comfortable sharing publicly. A newer computational approach consists of analyzing users' web browsing history (e.g., the news articles a user has clicked on) [17, 22–24, 31, 38]. While this approach addresses previously mentioned limitations, it still provides data only on a subset of news articles users are exposed to (i.e., the ones they subsequently click on). We tackle these limitations in this work by adopting a measurement methodology based on data donation, providing a comprehensive picture of the news landscape on social media.

## 2 METHODOLOGY AND DATASET

Our measurement methodology consists of building a non-intrusive tool enabling people to donate data about the news content they see on Facebook. In this section, we describe the design and technical considerations of the measurement infrastructure.

### 2.1 Monitoring tool

To enable users to donate data about the content they encounter, in a manner that does not disturb their regular activity, we implemented CheckMyNews, a privacy-preserving *browser extension* for Google Chrome that automatically collects, in the background, data about the content users see when browsing Facebook. The browser extension collects the following information from users:

i. *News posts:* CheckMyNews detects and collects posts related to news. To detect these posts, we compiled an extensive list of over 12,000 news outlets and the Facebook pages with which they are associated (see Section 2.2 for details on how we compile this list).

---

[1]https://anonymous.4open.science/r/CheckMyNews-AE8B

We consider a Facebook post as a *news post* if: (a) it was published by a page from our list of Facebook pages or (b) the landing URL of the post points to one of the news outlets we have in our list. A news post can be published or shared by the Facebook page of a news outlet or a random Facebook profile or page. A news post can be private (only a limited group of users can see it) or public (all users can see it). CheckMyNews collects the text, the media (e.g., image, video), the publisher, and the landing URL if the news post is public. On the contrary, if the news post is private, CheckMyNews only collects the landing URL and a hashed version of the publisher's username to keep it pseudonymous.

ii. *Other posts:* To evaluate the coverage of our list of news outlets and extend it, upon user permission, CheckMyNews can collect the *non-news public* and *non-news targeted posts* users see in their feeds, in addition to the news posts. This data is also useful to calculate the proportion of news posts among all the posts received by users, and compare how users interact with news posts vs. other posts. Note that CheckMyNews does not collect any private non-news posts (see Appendix A.1 for more details).

iii. *Visibility time of posts:* CheckMyNews collects how much time the post was visible on the user's screen for every post received on Facebook. For this, we check what post is visible on the user's screen every 0.5 seconds, and we start a time counter each time a new post becomes visible. The timer counts as long as more than 30% of the post is still visible.

iv. *Interactions with posts:* CheckMyNews collects both *visible* and *hidden* interactions of users with all the collected posts. The *visible* interactions are actions such as whether the user liked, disliked, commented, or shared a post. The *hidden* interactions are actions such as visiting the landing URL of the post, clicking on one of the images of the post, or checking who is the publisher of the post by visiting their Facebook profile. The hidden interactions are invisible to friends, while visible interactions are visible to friends.

v. *Survey data:* CheckMyNews has an option to send surveys to the user panel. It allowed us to request the participants to voluntarily disclose their demographic information, such as, gender and political affiliations (Democrat, Republican, or Independent).

CheckMyNews identifies users using a one-way hashed version of their Facebook IDs; their PIIs, such as names, usernames, or emails, are never sent to our servers. Overall, we have collected the news posts received by users on Facebook, their visibility time, and how users interact with them. To our knowledge, none of the prior works have looked at such granular information. This is partly due to the difficulty of collecting such data as Facebook changes its HTML markup, sometimes adversarially, to disturb data collections from tools such as ours [26]. We devoted considerable time to developing a monitoring tool that can enable reliable data donation (see Appendix A.2 for more details).

## 2.2 Extended lists of news outlets

The list of news outlets we monitor determines the breadth of our view of news posts that users are exposed to and consume. To have a comprehensive view, *we need an extensive list of news outlets to monitor.* However, most previous studies have only monitored a limited number of traditional news media [23, 30, 45], and the research community lacks a comprehensive list of news outlets

active on social media. To overcome the limitation, we employ three approaches.

*First*, we rely on News Guard and Media Bias Fact Check, two independent data providers that survey news outlets and provide qualitative information about them (e.g., political leaning, quality). News Guard contains 2,939 news sites, while Media Bias Fact Check contains 2,062 news sites. We call the aggregate list of 4,149 news outlets the Established News Sites list. For every news outlet, we also collected the corresponding Facebook page. Overall, we have a list of 4,323 Facebook pages corresponding to the established news domains, which we call Established Facebook Pages list.

We know from recent reports that there is an emergence of sites that claim to be news organizations, especially before elections [42]. Hence, it is essential to go beyond established media sites and consider news outlets that simply *claim* to be news media, irrespective of their reputation, popularity, or whether they create original content or are simply content farms. We refer to these sites as *under-the-radar* news outlets, and we hypothesize that some use Facebook to advertise their content, as they probably have a small organic reach. Hence, our *second* approach consists of grabbing from the Facebook Political Ad Library [16] all Facebook pages that promoted a political ad in the U.S. between June 2018 and June 2020 and claim to be "News Media." Then, for every Facebook page, we extract the website mentioned in the "About" page. Although not all Facebook pages mention a website, we could gather 8,084 websites for 9,679 different Facebook pages. Hence, this method provides us with a significant number of sites claiming to be news outlets that News Guard and Media Bias Fact Check have not reviewed.

*Third*, as mentioned earlier, CheckMyNews can collect the targeted and public posts on users' feeds. Most of these posts come from Facebook pages. Hence, we select all Facebook pages that claim to be "News Media" in their "About" section and extract the news domain if they mention it. This method gave us 404 additional news domains associated with 449 Facebook pages.

In total, our list of under-the-radar news media sites (collected with the second and third methods) contains 8,489 different domains associated with 10,128 different Facebook pages. We call the corresponding lists the Under-the-Radar News Sites list and the Under-the-Radar Facebook Pages list. Overall, our list of news media websites contains 12,638 different news domains and 14,451 Facebook pages. To the best of our knowledge, this is the most extensive list of (both established and under-the-radar) news domains in the US.[2]

## 2.3 Ethical considerations

CheckMyNews collects sensitive and personal data from the study's participants. Before collecting any data, we obtained the necessary approvals from the Data Protection Officers and the Ethical Review Board of our institution and the participants' explicit consent. Additionally, we use various strategies to minimize security and privacy risks for the study's participants, and we comply with the EU General Data Protection Regulation (GDPR). We provide more details about our ethical considerations and security and privacy risk minimization strategies in Appendix A.3.

---

[2]Downloadable at anonymous.4open.science/r/US_News_Outlets_Dataset-342F

## 2.4 User recruitment and dataset

We posted about the tool and the surrounding concept of data donation on Prolific, and 889 users living in the U.S. agreed to install CheckMyNews on the computer they use to connect to Facebook and keep it active for six weeks (between November 2, 2020, and February 15, 2021, centered around the U.S. presidential elections). While we can not guarantee representativeness, we tried to reach out to Prolific users across various U.S. states and ethnicities. We provide a detailed breakdown of age, location, ethnicity, and political affiliation as reported by the users in Appendix B.2.

Some users dropped out from the study and uninstalled the extension in between (see Appendix B.1). Hence, our analysis focuses on data from 472 users who spent at least 30 minutes browsing Facebook or received more than ten news posts in their feeds. We found that all 472 users had typical browsing activity on Facebook, with a reasonable number of posts with respect to the time spent on Facebook. Hence, it is unlikely that any of these users are bots browsing on Facebook (see Appendix C.2).

Our dataset contains 143,129 news posts. For each user, on average, we collected 303 news posts ($M(median) = 57, s(std) = 789$) and 6 news posts per active day ($M = 3, s = 10$). Moreover, we have collected a total of 8,612 user interactions with 5,386 different news posts. Each user interacted on average with 18 posts ($M = 2, s = 60$). (see Appendix C.1 for more statistics).

## 2.5 Limitations

Despite our best efforts, our data collection methodology has two main limitations. *First*, our monitoring tool cannot capture news exposure and consumption on users' mobile phones. Sadly, providing such a tool for mobile phones is technically very challenging. There is currently no data suggesting that Facebook news diets on mobile phones significantly differ from news diets on web browsers in terms of composition, quality, and diversity of news posts. *Second*, since the study requires the collection of personal data from users, understandably, many users are uncomfortable with donating data, and thus, they would be reluctant to install a monitoring tool. This makes it challenging to obtain a large and representative sample of users. Despite the limitations, we believe that the compiled dataset goes much beyond previously compiled datasets in terms of comprehensiveness and detail, and provides much-needed answers to several long-standing questions in the community.

## 3 WHAT NEWS DO PEOPLE GET ON SOCIAL MEDIA?

This section attempts to answer longstanding questions about the exposure and consumption of news that appear on users' social media feeds. We answer these questions by examining the underlying mechanisms responsible for news appearing in users' feeds.

## 3.1 Types of Social Media News Exposure

Prior works have looked at exposure to news in social media in silos. Few works have focused on incidental exposure [19, 45, 53], while others on selective exposure [8]. For instance, [19, 53] considered that incidental exposure consists of all news users encounter on search engines or social media platforms when they use them for a purpose different than seeking news; whereas [8, 33] considered

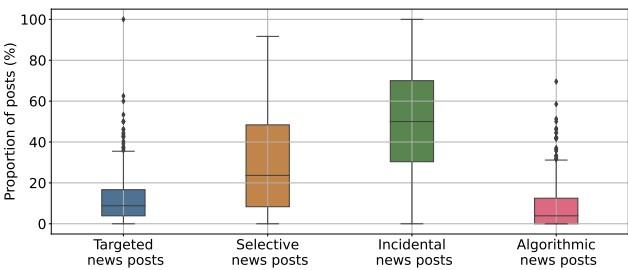

**Figure 1: The distribution of the proportion of selective, incidental, targeted, and algorithmic news posts received by each user in our dataset.**

that all tweets a user potentially receives are selective news. Both interpretations are justifiable, even if they seem contradictory.

We argue, however, that we need to differentiate social media exposure based on the *underlying mechanisms* responsible for the news appearing in users' feeds, especially if we want to characterize the consumption pattern and the quality and diversity of news. Such a systematic perspective allows a better understanding of root causes and finding potentially better technological and algorithmic designs. We identify the following four mechanisms through which news posts (and posts in general) can appear on users' feeds.

**A. Selective news exposure**: On Facebook, users can follow or like the pages of their preferred news media sites. This enables them to receive posts published by these pages in their feeds. If a user no longer wants to receive such content from a particular Facebook page, they can unfollow it. Hence, the user controls which posts from which news outlets appear in their feeds – this is a form of selective news exposure.

**B. Targeted news exposure**: Advertisers can pay the Facebook ad platform to show specific content in the feeds of users who satisfy distinct characteristics (i.e., targeting criteria). While most of the ads users see are for commercial products, these targeting mechanisms are increasingly being used for political propaganda [25] and propagating news items. For example, an advertiser can pay Facebook to show ads containing links to a specific news article to an audience "interested in climate change, living in San Francisco." These news posts appear in users' feeds because advertisers want them. Advertisers on Facebook can utilize $250K+$ attributes to define their audiences, which allows them to expose people with very precise interests to particular news stories [1].

**C. Algorithmic news exposure**: Facebook users receive "Suggested for You" posts on their feeds. These are personalized posts that Facebook's algorithms determine to be relevant for users, primarily based on their previous engagement and behavior on Facebook [15]. Note that these posts are neither shared by friends nor paid by advertisers. While not all "Suggested for You" are news-related, a fraction of them could be.

**D. Incidental news exposure**: Finally, all other posts with links to news articles, which are not targeted and do not result from a user following a news page, form incidental news exposure. It includes (a) posts from friends, groups, or pages that either directly share or re-share posts with a link to a news article and (b) posts from news sources' Facebook pages that share links to other news sites.

Appendix C.3 describes in more detail how we technically identify the four types of exposure in our dataset.

The first elemental question we ask is *what proportion of news exposure on social media is selective, incidental, algorithmic, or targeted.* Figure 1 shows the distribution of the proportion of selective, incidental, algorithmic, and targeted news posts received by users. Median values across all users show that a user's Facebook feed contains 50% of incidental, 24% of selective, 4% of algorithmic, and 9% of targeted news posts. It is noteworthy that only 38%, 13%, and 11% of users did not receive algorithmic, targeted, and selective news during the study period. Hence, most users follow different news providers; they are targeted with news by advertisers and are exposed to algorithmic news on Facebook, resulting in all four types of exposure being prevalent across users.

Moreover, we find that the ratio of targeted, selective, incidental, and algorithmic news posts varies with time (see Figure 8 in Appendix C). Precisely, we find that targeted news posts reached their highest proportion at two different periods: (i) at the beginning of November, coinciding with the general election day on November 3*rd*, and (ii) at the end of December 2020, coinciding with the vote of the Electoral College members on December 14*th* [11]. Hence, the study participants were targeted with more news advertisements during these two sensitive periods.

These results indicate that major offline events might directly impact the composition of news on social media, emphasizing the need to distinguish and analyze each type of exposure separately. Precisely, the fact that the users were targeted more around two important dates during the U.S. elections is particularly alarming considering that some news publishers are exempt from the Facebook ad authorization process when targeting U.S. users with political advertisements, and their ads are not labeled as related to politics and are not listed in the Facebook Ad Library [14]. This opens up the possibility of voter manipulation through targeted advertisements in a stealth mode, evading scrutiny.

## 3.2 Quality of Social Media News

This section assesses the quality of news users receive on Facebook and how prevalent misinformation is on their feeds. We characterize the quality of a news post at the source level, i.e., we consider a post to have the same quality as the media source publishing the news. Recall that we collect posts from both news organizations reviewed by Media Bias Fact Check and News Guard (Established News Sites list) and news from under-the-radar websites associated with Facebook pages claiming to be news providers (Under-the-Radar News Sites list). Based on the available information, we use three indicators to evaluate the quality of a news source:

(a) Whether it is considered as repeatedly spreading *misinformation* and conspiracy theories by either Media Bias Fact Check or News Guard. Out of the 4,149 domains in the Established News Sites list, 456 (11%) are considered low-quality (i.e., repeatedly spreading misinformation). These domains are associated with 467 Facebook pages (see Appendix C.4 for details on the misinformation ratings provided by Media Bias Fact Check and News Guard).
(b) Whether it is an *under-the-radar* news source not covered by Media Bias Fact Check and News Guard. While not all under-the-radar sites necessarily spread low-quality information, we know

| | All news posts | Targeted news posts | Selective news posts | Incidental news posts | Algorithmic news posts |
|---|---|---|---|---|---|
| **Factual news sources** | 63% | 52% | 64% | 66% | 64% |
| **Misinformation news sources** | 5.1% | 2.5% | 5.8% | 5.0% | 4.5% |
| **Under-the-radar news sources** | 13% | 32% | 16% | 7% | 12% |

**Table 1: Fraction of posts from factual, misinformation, and under-the-radar news sources across all users. The table excludes posts from under-the-radar sources and posts from established sources rated as Mixed.**

that advocacy groups have created under-the-radar news sources shortly before the U.S. elections to spread information (or misinformation) [43]. Hence, we consider them as potentially suspicious. (c) Whether it is considered as spreading mostly *factual* information by Media Bias Fact Check or News Guard. There are 2,942 such news sources, that correspond to 3,074 Facebook pages (see Appendix C.4 for more details on the factualness ratings provided by Media Bias Fact Check and News Guard).

There are 723 domains in the Established News Sites list that are considered to be sharing mixed content (factual and misinformation) and 28 domains for which Media Bias Fact Check and News Guard do not have a quality evaluation.

Table 1 represents the fraction of posts from factual, misinformation, and under-the-radar sources across all users. It shows that 5.1% of news posts users are exposed to are from news sources known for repeatedly spreading misinformation or conspiracy theories. Additionally, we observe that selective exposure has the highest rate of posts from these low-quality sources (5.8%), followed by incidental (5.0%), algorithmic (4.5%), and targeted exposure (2.5%). The differences are statistically significant (Pearson's chi-squared [40]; $p < 0.001$), and hence, it seems that users are more likely to expose themselves to sources known to spread misinformation than be exposed through their communities (i.e., incidental exposure) or platform's algorithms (i.e., algorithmic exposure). This is a new and intriguing observation.

Despite the prevalence of misinformation, on average, 63% of news users receive are from factual news sources. Incidental, selective, and algorithmic exposures have higher rates of posts from mostly factual sources (66%, 64%, and 64%, respectively) compared to targeted news exposure (52%). On the other hand, 13% of news received by users are published by under-the-radar Facebook pages or have URLs to under-the-radar news sites. Targeted exposure has the highest rate of posts from under-the-radar news sources (32%), followed by selective (16%), algorithmic (12%), and incidental exposure (7%). This indicates that targeted advertising is the main driver that exposes users to under-the-radar content and may represent a threat to the quality of news diets.

## 3.3 Diversity of Social Media News

An important question that has captured attention is whether users' online news exposure is politically diverse. Prior works have presented conflicting evidence, uncovering both the presence and absence of so-called "filter bubbles" [35, 45]. We revisit this question

| All news posts | Targeted news posts | Selective news posts | Incidental news posts | Algorithmic news posts |
|---|---|---|---|---|
| 83.8% | 70.7% | 47.0% | 81.5% | 71.0% |

**Table 2: Fraction of users with diverse news diets, who were exposed to posts from both left and right-biased sources.**

armed with information about *actual* news exposure of users, in contrast to prior works' reliance on approximations: user surveys [39], web browsing histories [5, 45], or the social network (e.g., tweets from accounts followed by a user) [8].

We attempt to provide a realistic diversity landscape as our data collection includes (i) a complete and precise list of the news posts users have received/seen on Facebook, and (ii) reliable labeling for news sources' political leanings provided by Media Bias Fact Check and News Guard (see Appendix C.4 for details). 86% of all news posts in our dataset were published by or with landing URLs to established news sources having the corresponding political leaning labels. We only consider these posts for our analyses. Furthermore, for each category of news posts, we only consider users who have received at least 10 news posts, resulting in 232, 351, 140, and 145 users for selective, incidental, targeted, and algorithmic news posts respectively (414 users if we consider all categories).

Similar to [18], we use two different metrics to measure political diversity. The first metric is **diversity** which captures whether a user has been exposed to news sources from both sides of the political spectrum (left and right). It is a binary metric, taking the value of 1 if a user received at least one post from a left-leaning news source and at least one post from a right-leaning news source. The second metric, termed **balance**, focuses on the ratio of left vs. right-leaning sources (or right vs. left-leaning sources, if the latter is higher) that a user encounters. It is a measure of the proportion of posts from left sources compared to right sources (or vice versa) to which a user was exposed. It varies between 0 and 1, where 0 signifies that a user did not receive any posts from one of the political leanings (left or right), 0.5 indicates that a user received twice as many posts from one leaning vs. another, and 1 indicates that a user received an equal number of posts from both sides.

We calculate the values for these two metrics for each user, considering all categories of news posts together as well as separately. Table 2 presents the proportion of users with diverse news feeds. We observe that 83.8% of users received at least one post from both sides of the political spectrum. When considering targeted, incidental, and algorithmic news posts separately, most users have received politically diverse news for these three categories. However, when it comes to selective news posts, more than half of the users did not encounter even a single story from the other side. These results indicate that users mainly subscribe to news sources of the same political leaning (selective exposure) but get exposed to sources from the opposite side of the political spectrum through incidental, targeted, and algorithmic news exposure.

Figure 2 presents the diversity of news exposure captured through the *balance* metric. We observe that algorithmic and targeted news diets are the most balanced ones, followed by incidental news diets, while selective news diets are the least balanced ones. If we consider a news diet to be *well-balanced* when the balance metric ≥ 0.5 (i.e., at least one-third of posts are from each ideological

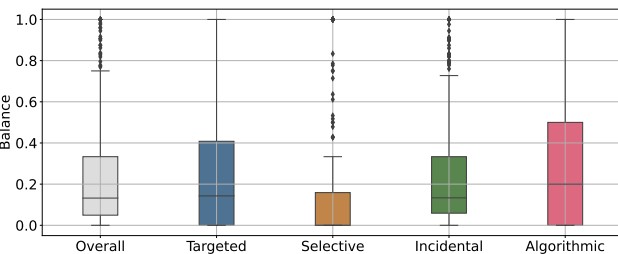

**Figure 2: Distribution of balance: fraction of news posts from left vs. right-leaning sources (or right vs. left if the latter is higher) per user. Value 0 represents users who did not receive any posts from one of the political leanings (left or right); 0.2 represents users who received $\frac{1}{5}$th posts from one political leaning compared to the other; while a value of 1 indicates an equal number of posts from left and right news sources.**

leaning), we find that 28% and 22% of users have well-balanced algorithmic and targeted news diets, while only 17% and 13% of users have well-balanced incidental and selective news diets. These observations are consistent with "diversity" results supporting that algorithmic, targeted, and incidental news exposure leads users to more balanced news diets than selective exposure alone.

## 3.4 Consumption of Social Media News

CheckMyNews captures how users interact with news posts, including the time a post was visible on users' screens, whether they read the corresponding news article, checked the publisher's page, and commented, liked, or shared it with their friends.

*3.4.1 Visible vs. hidden interactions.* There are two types of interactions: visible and hidden. *Visible interactions* are actions visible to other Facebook users (including a user's friends), such as commenting, sharing, or liking a post. *Hidden interactions* are interactions that are not visible to a user's friends, such as clicking on the post to visit the actual article, clicking on the page of the publisher, clicking on the image of the post, saving, reporting, or hiding the post. We make this distinction because users might behave differently when their actions are visible to their friends (e.g., they might click on a news post but may not want their friends to know). Interestingly, *we find that users performed visible interactions on only* 2.6% *and hidden interactions on only* 2.8% *of the news posts they received* – a tiny minority of news they get exposed to on Facebook. We further see that users performed both hidden and visible interactions on < 0.5% of posts, suggesting that the visible and hidden interactions are performed mostly on different sets of posts. In fact, we find that users accessed the landing URLs of only 14% of the news posts they have shared. It is a very surprising finding hitherto unreported in any prior work.

Another essential information CheckMyNews captured is the time a particular news post was visible on the user's screen. To account for cases where a user has stopped scrolling and moved away while a Facebook post is visible on their screen, we use the interquartile range to detect large values and exclude them from the analysis. In addition, CheckMyNews collects the visibility times of non-news content to use them as a reference point. We classify

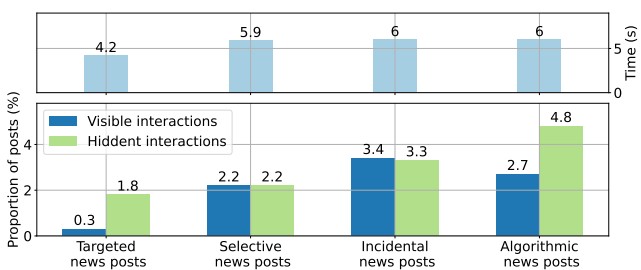

**Selective, incidental, algorithmic and targeted news posts**

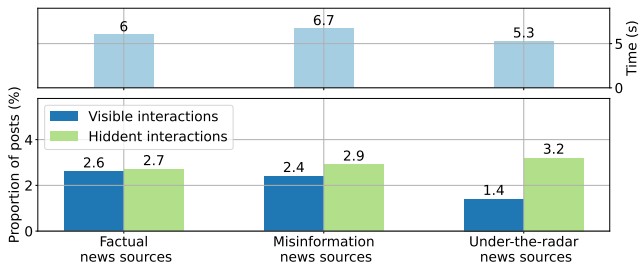

**Posts from factual, misinformation and under-the-radar sources**

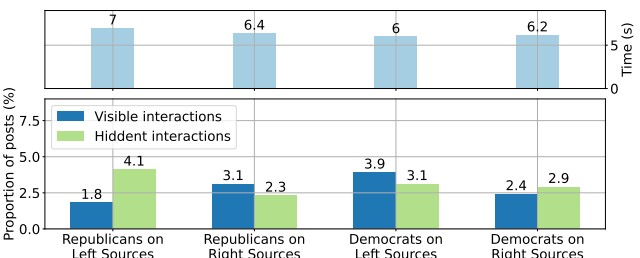

**Posts from left/right sources received by democrats and republicans**

**Figure 3: Fraction of news posts on which users make visible and hidden interactions as well as the median visibility time of news posts on user's screens.**

non-news content into non-news ads (i.e., posts with a "Sponsored" tag) and non-news posts (from friends or communities). *We find that the visibility time of news posts on user's screens (median 5.6 seconds) is higher compared to the visibility time of non-news ads (median 4.0 seconds) and non-news posts (median 4.1 seconds) (Kolmogorov-Smirnov test [7]; $p < 0.001$).* While we can quantitatively measure how long a post was visible, we do not know if users spend more time on news posts because they find them more interesting or if the cognitive load of reading a news post is higher than the cognitive load of other posts. Nevertheless, even though users do not click or react to most news posts, they do see them for an amount of time longer than other posts. Hence, this raises an important question for future work: *to what extent does reading a news post without going to the landing URL affect a user's beliefs and knowledge, and how long do users need to look at a post to remember it?*

*3.4.2 Effect of the type of exposure.* We next investigate whether users interact differently with selective, incidental, algorithmic, and targeted news posts. Figure 3 shows the fraction of news posts on

which users make visible and hidden interactions as well as the median visibility time of news posts on user's screens. The figure shows significant differences in how users interact with selective, incidental, algorithmic, and targeted news posts (Pearson's chi-squared; $p < 0.001$ for both visible and hidden interactions, and Kolmogorov-Smirnov; $p < 0.001$ for visibility time). While the fraction of visible and hidden interactions is similar for selective and incidental news posts; the fraction of hidden interactions is 1.8 to 6 times higher than visible interactions on algorithmic and targeted news posts. It may be possible that users perceive the underlying mechanisms through which the news posts appear in their feeds differently, and this might inhibit visible interactions on algorithmic and targeted news posts (e.g., users might avoid sharing an article that was recommended to them by Facebook's algorithms, while they will share more freely an article that comes from their friends). We leave a causal validation of this hypothesis through a randomized controlled trial as future work.

*3.4.3 Impact of news source quality.* Next, we investigate whether users interact differently with news posts from sources of varying quality. We split news posts into three categories: (a) news posts from the ESTABLISHED NEWS SITES LIST that News Guard and Media Bias Fact Check consider to be factual, (b) news posts from the ESTABLISHED NEWS SITES LIST that these two agencies consider to be repeatedly spreading misinformation, and (c) news posts from the UNDER-THE-RADAR NEWS SITES LIST.

Figure 3 presents the consumption statistics for factual, misinformation, and under-the-radar news posts. The figure shows that news posts from misinformation news sources have a longer visibility time on users' screens (median: 6.7 seconds) than news posts from factual news sources (median: 6.0 seconds) (Kolmogorov-Smirnov; p < 0.001). *This result is intriguing and raises questions on whether a different cognitive process gets triggered when faced with misinformation compared to factual news sources (e.g., they spend more time to be sure about the factuality).* We leave the analysis of the root cause of this observation as future work.

Additionally, users generally pay less attention to news posts coming from under-the-radar news sources (median: 5.3 seconds). Furthermore, while the fraction of visible and hidden interactions is not significantly different for posts from factual and misinformation sources, the fraction of visible interactions on under-the-radar posts is significantly lower than the fraction of hidden interactions.

*3.4.4 Impact of concurrence in political ideology.* Finally, we investigate how users interact with posts with similar or opposite political leaning. In our survey, we asked users about their political leaning (Republican, Lean Republican, Independent, Lean Democrat, Democrat, or Other). We consider four sets:
(a) posts from right-leaning sources received by Democrats,[3]
(b) posts from right-leaning sources received by Republicans,[4]
(c) posts from left-leaning sources received by Democrats, and
(d) posts from left-leaning sources received by Republicans.

Figure 3 shows the consumption behavior. When considering viewing time, we do not observe statistically significant differences between these four scenarios, however, the difference begins to

[3]Users who self-identify as Democrat or Lean Democrat.
[4]Users who self-identify as Republican or Lean Republican.

emerge when we consider visible and hidden interactions (Pearson's chi-squared; p < 0.001). The figure expectedly shows that users perform more visible interactions on posts published by sources with a matching political ideology (Republicans interact with 3.1% of posts from right-leaning sources vs. 1.8% from left-leaning sources; Democrats interact with 3.9% of posts from left-leaning sources vs. 3.1% from right-leaning sources). However, when it comes to hidden interactions, we observe that users interact more with posts from sources with the opposite political ideology (Republicans interact with 4.1% of posts from left sources vs. 2.3% from right sources; Democrats interact with 2.9% of posts from right sources vs. 2.4% from left sources). *This provides a hopeful insight into users' online behavior, which attests to their willingness to engage with opposing views, albeit in stealth mode.*

## 4 CONCLUDING DISCUSSION

In this work, we attempted to provide a realistic view of users' news exposure and consumption on Facebook. Our methodology consists of building a non-invasive monitoring tool that can allow a large number of Facebook users to donate data on the news posts they receive on Facebook and how they interact with them. We could provide a *realistic*, *fine-grained*, and *broad* representation of users' exposure to news and their consumption behavior by capturing the actual news users are exposed to on Facebook and precisely what users see on their screens. Overall, our measurement approach provided immensely valuable data on news exposure that only online platforms have had access to until now and no outside researcher could have the opportunity to avail.

Of course, implementing such measurement infrastructure is technically challenging. First, we had to ensure that the data collection was reliable and did not miss data due to the variability in how online platforms serve their content. Second, we had to ensure the privacy and safety of the data collection and storage. Our codebase is publicly available[5] to help other groups adopt such methodology and encourage developing infrastructure to study news exposure and consumption on other social media platforms [55].

**Implications of results.** Our research offers a unique opportunity to delve into users' precise exposure and engagement with news-related content, allowing us to reliably measure the prevalence of misinformation and the political diversity within Facebook news diets. We highlight three noteworthy aspects of our findings:

*Mechanistic perspective on news exposure and consumption.* We focused on investigating news exposure and consumption patterns, specifically considering how news articles appear in users' feeds based on the underlying mechanisms. We found statistically significant distinctions in diversity, quality, and consumption behavior within four exposure categories: selective, algorithmic, incidental, and targeted. For instance, we observed that incidental exposure leads to more balanced and factual news consumption, while users are less inclined to share targeted news posts. These differences highlight the importance of adopting a mechanistic lens when attempting to model and understand dissemination and consumption of news on social media platforms.

*Transparency for targeted news exposure.* Recent research works have shown a significant *shift* from using targeted advertising as a

way to promote products to a way to promote information [2, 47, 52]. For the subcategory of ads with political messages, such powerful targeting is now being regarded as a danger, and many lawmakers are proposing to regulate such practices through increased transparency and targeting restrictions (e.g., Digital Services Act [13], European Democracy Action Plan [12]). However, news organizations are currently exempted from such obligations and restrictions. We believe that malicious actors can easily leverage AI-driven targeting technologies to promote news-related posts that *leverage user's data to deceive and manipulate them*, by targeting news that resonate with each user [27, 28, 44, 46, 51]. Our results show that exposure to targeted news is a general phenomenon and represents an important fraction of users' news exposure. Therefore, we believe that the same levels of transparency for targeted news should be imposed similar to political advertising.

*News consumption behavior.* Our study uncovered a number of hitherto unknown and intriguing news consumption patterns: users engage significantly less with targeted and algorithmic news-related posts; they tend to spend more time reading content from misinformation sources compared to factual news sources; and share news-related posts without reading the actual articles. We also found evidence that users engage with posts from sources with opposing political ideologies through hidden interactions.

**Scaling data donation.** Installing monitoring tools on personal computers can be unsettling for many users. It's understandable, as such tools could potentially exploit users' trust, jeopardizing their security and privacy. To address this concern, we take a transparent approach by making our code publicly available for auditing by anyone. Nonetheless, it is essential to emphasize the significance of data donations in uncovering risks with current technologies and promoting this practice more widely. While we acknowledge that achieving perfect representativeness may be challenging, we made efforts to reach out to Prolific users from diverse backgrounds, including various U.S. states and ethnicities. We are aware that the presence of a monitoring tool on users' computers might influence their behavior, potentially leading to altered usage patterns. This issue is not uncommon and has been encountered in previous research analyzing browsing histories. However, by requesting users to keep the tool installed for an extended period, we anticipate that their behavior will stabilize over time.

**Future work.** We believe that our paper can trigger a plethora of future research works seeking to understand the underlying factors and implications of these consumption behaviors. Some of the potential research questions include: Does targeted and algorithmic exposure exert a lesser impact on users' opinions compared to selective and incidental exposure? Can the amount of time spent on news posts from misinformation sources provide insights into whether users question the information presented? To what extent does merely seeing a news post without reading the source article influence the user's opinion on a particular topic? Do interactions with posts from sources holding opposing political ideologies reinforce or prompt users to question their own political preferences? Addressing these questions could shed further light on the complexities of news consumption patterns on social media, the role of exposure mechanisms, and their effects on users' opinions.

---

[5]Available at https://anonymous.4open.science/r/CheckMyNews-AE8B

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

# A  MEASUREMENT METHODOLOGY

## A.1  Types of Facebook posts collected

We collect information about three types of user posts on Facebook:

1. *News posts* – news posts (or news-related posts) include Facebook posts that fall into two categories. First, posts published by pages listed in our list of Facebook pages associated with news sources (Established Facebook Pages list or Under-the-Radar Facebook Pages list), and second, posts that contain a landing URL directing to one of the news domains in our lists (Established News Sites list or Under-the-Radar News Sites list). These posts can be targeted (i.e., ads), suggested by Facebook, and have different privacy settings, ranging from private (visible to a limited group of users) to public (visible to all users).

2. *Non-news targeted posts* – Posts advertisers pay Facebook to send to specific groups of users [2]. Note that CheckMyNews detects the news posts before detecting the targeted ones. Hence, all the posts it collects as targeted posts are not news-related.

3. *Non-news public posts* – All non-targeted and non-news posts. They are shared by users, groups, or Facebook pages and set as public by the publishers, making them visible to anyone.

While non-news targeted and public posts do not meet our criteria for being news-related based on the filter we applied during the data collection, their inclusion was valuable in expanding our lists of news domains and their Facebook pages. By analyzing the Facebook pages from which we collected these public and targeted posts, we identified 404 additional news domains associated with 449 Facebook pages with the "News Media" category.

## A.2  Reliability of the monitoring tool

Having a *reliable monitoring tool* that does not miss any post a user sees on Facebook is necessary to have correct and coherent measurements. To ensure the tool works well for all users during the data collection, we have implemented several tests at the extension level to detect when our collection functions do not work correctly (e.g., the user is on Facebook, but we do not detect any post for more than 120 seconds). We send error messages to the server, and we have developed a monitoring page that we consult daily to check for aggregate and per-user statistics and consult the error messages. The targeted posts are more challenging to detect because Facebook renders them using complex changing HTML objects that sometimes differ between users. To cope with this, we first make sure that the targeted posts we miss are collected as public posts (that are simpler to detect). We then manually check users from whom we have collected only public posts and investigate how targeted ones are rendered for them. We finally updated our extension to detect the targeted posts rendered in this new way.

Overall, the monitoring tool can easily be installed by users, works silently in the background, has a minimal impact on browser performance, and does not affect the user experience.

## A.3  Compliance with ethical principles

We use various strategies to minimize user security and privacy risks and sought the necessary approvals from Data Protection Officers and Ethical Review Boards. The personal data we collect is handled following the EU General Data Protection Regulation 2016/679. Personal data is processed lawfully, fairly, and in a transparent manner. To ensure privacy, confidentiality, security, and legality, we took the following measures:

(a) *Data minimization*: The tool collects information about the content users *receive* and not the content they share. Additionally, we only collect the landing URLs or private *news* posts.

(b) *Pseudonymization:* We do not send our servers any personally identifiable information of users (e.g., email, name, phone number). No summary data is disclosed that would allow inference about an individual's personal or private data. Each user is identified by a random identifier generated at each new tool installation.

(c) *Explicit consent:* Every user installing our tool is shown a page describing precisely the data given and the use of this data. We ask for the user's explicit consent to donate data and participate in the research study. The consent form is submitted (electronically) for each user installing the tool, and we keep proof of this consent.

(d) *Detailed privacy and security risks assessment:* We passed a security homologation from our institution and wrote a detailed document that analyzes security and privacy risks at every level of the data transfer and worked with network and system engineers from our University to secure the application at every level.

(e) To use our tool, users must confirm being at least 16 years old.

(f) *Data removal/leaving the study:* We informed the participants of their right to access, correct, request portability, and delete personal data, and we gave them the contact details of our Data Protection Officer (DPO) to exercise their rights. Participants could leave the study at any moment and ask for their data to be removed.

# B  RECRUITING AND REPRESENTATIVENESS

## B.1  User recruiting

We posted about our study on Prolific and 889 U.S.-based participants agreed to install CheckMyNews and keep it active for six

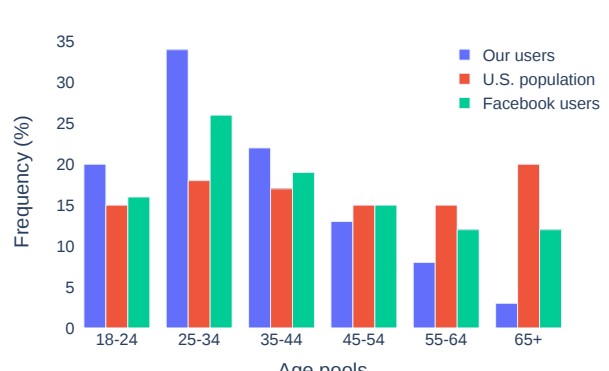

Figure 4: Age distribution of our users vs. U.S. population and U.S. Facebook users.

weeks (between November 2020 and February 2021). To compensate the participants for their time in installing and answering the survey questions, we offered them an initial payment right after the installation and a bonus payment at the end of the six weeks if there was a minimum required activity level. Only 720 successfully installed the extension, and only 580 logged into Facebook after installing it. Finally, only 472 users kept the tool active for a long period and respected the minimum Facebook activity condition (at least 30 minutes); we do our analysis only on these 472 users. We consider that users have dropped out of our study when we stop collecting their activity. We do not know whether they have uninstalled/disabled the browser extension or stopped using the computer or the browser on which they have installed it. Though the initial study was launched over six weeks, we extended our dataset to include data over three months (until February 15, 2021) since we had many users who kept running the extension.

## B.2   User representativeness

Our users are 65% males and 35% females (compared to 45% males 55% females for U.S. users on Facebook [48], and 49% and 51% for the U.S. population [50]) and live across 48 states in the U.S. The users are part of different ethnic groups: 74% White, 11% African American, and 11% Asian (compared to 76%, 13% and 6% for the U.S population [49]). Figure 4 presents the age distribution of our users, compared to the overall U.S. population [41] and the Facebook U.S. users [2]. More than half of them are between 20 and 40 (54%). Hence, our database has more young users than the normal U.S. population, but we have users of all age pools. According to the survey, 76% of users consider themselves Democrats, while 16% are Republicans and 6% are Independents.

## C   DATA COLLECTION AND PROCESSING

### C.1   News exposure and consumption data

Our dataset includes 889,438 Facebook posts received by 472 users; 143,129 (16%) posts are news-related, 205,469 (23%) are non-news targeted and 548,152 (61%) are non-news public. Out of all the news-related posts, 108,659 posts have a link to one of the news domains in our lists (ESTABLISHED NEWS SITES list or UNDER-THE-RADAR NEWS SITES list) and 85,066 were published by pages in our

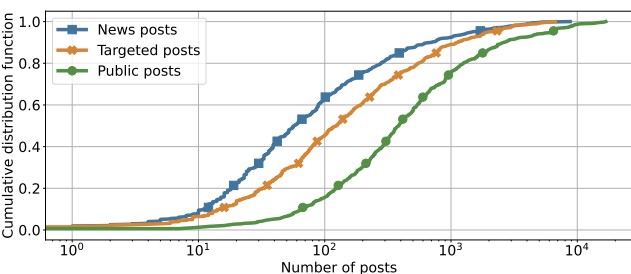

Figure 5: Number of news posts, targeted posts and public posts across all users.

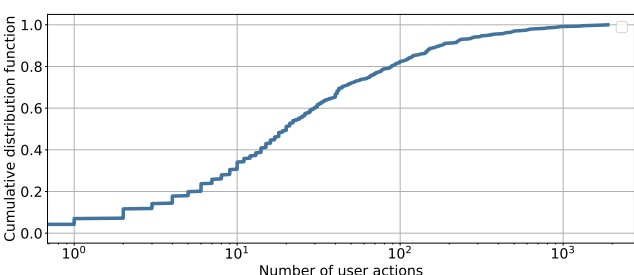

Figure 6: Number of interactions with news posts by user.

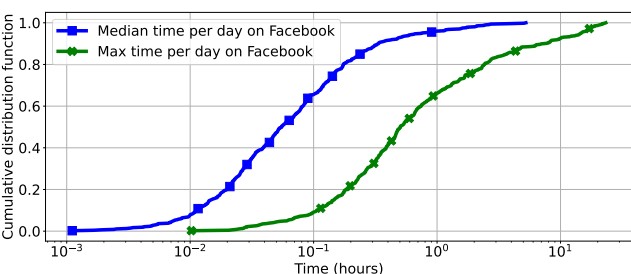

Figure 7: CDF of the maximum and the median time spent on Facebook per day per user.

lists of Facebook pages associated with news media (ESTABLISHED FACEBOOK PAGES list or UNDER-THE-RADAR FACEBOOK PAGES list). Figure 5 represents the number of news posts, targeted posts, and public posts received by users. A median user received 125 ($\overline{x}$ = 435, $s$ = 855) targeted posts, 58 ($\overline{x}$ = 303, $s$ = 789) news posts, and 387 ($\overline{x}$ = 1,161, $s$ = 2,266) public posts. We have also collected a total of 37,300 user interactions with 24,486 different posts: 9,804 are visible interactions while 27,496 are hidden interactions. Figure 6 presents the number of actions performed by each user on all news posts. We can see that a median user performed 20 interactions ($\overline{x}$ = 79, $s$ = 189).

### C.2   Unexpected user behavior and bot detection

Considering the relevancy of the research questions we address in this study, it is crucial to ensure that none of the participants used bots during data collection. Figure 7 presents the median and maximum times spent on Facebook per day, over all active days for each of the 472 participants. The figure shows that an average user spent 0.06 hours (3.6 minutes) on Facebook on a day with median activity and 0.53 hours (32 minutes) on the most active

day. Furthermore, the figure reveals that 10 users spent more than 7 hours on Facebook on their busiest days. Upon investigating the posts collected from these users, we found no evidence of bot activity. Our analysis suggests that these users have left Facebook open on their browsers without actively browsing on the platform.

### C.3 Distinguishing news posts categories

Our monitoring tool collects all news posts on users' Facebook feeds. This section presents how we technically divide these posts into targeted, selective, incidental, and algorithmic news posts.

*Selective news exposure*: We select posts originating from the official Facebook pages of news media sites (Established Facebook Pages list and Under-the-Radar Facebook Pages list). We then check whether these posts contain a landing URL that directs users to their respective news media website. For instance, if the Facebook page of CNN–https://www.facebook.com/cnn–publishes a post that links to an article on cnn.com–https://edition.cnn.com/...), we consider it as selective exposure. However, when a news media Facebook page shares a post containing a link to a news article from an external site, we do not consider it selective exposure. In such cases, the user's exposure to the external site's content does not result from their explicit following of the external site's Facebook page.

*Targeted news posts*: This category includes all targeted posts that promote articles from news media sites, irrespective of the Facebook page that promotes them. While such posts are rendered similarly to regular Facebook posts, they include a "Sponsored" tag. We use several HTML and CSS selectors to identify this tag.

*Algorithmic news posts*: This category includes Facebook news-related posts that Facebook suggests to users. Such posts have the "Suggested for you" tag that we detect using CSS selectors. We analyze the HTML objects of all identified news-related posts and consider algorithmic exposure all news posts that include this tag.

*Incidental news posts*: For each news post, we extract the landing domain and the Facebook page's ID. We then verify if one of the following conditions is met: (a) the page's ID is not in our list of Facebook pages of news sites (Established Facebook Pages list and Under-the-Radar Facebook Pages list), but the landing domain is among our list of news sources (Established News Sites list and Under-the-Radar News Sites list), or (b) both the page's ID and the landing domain are present in the respective lists, but the Facebook page belongs to another source.

Our dataset contains a total of 143,129 news posts; 62,434 are selective, 60,529 are incidental, 11,566 are targeted, and 8,600 are algorithmic. Figure 8 illustrates the changing proportion of incidental, selective, targeted, and algorithmic news posts by week, during our data collection period.

### C.4 Metadata on news outlets

To assess the quality of Facebook news diets, we measure the proportion of posts originating from (a) mostly factual news sources and (b) sources spreading misinformation, fake news, and conspiracy theories. We also evaluate the political diversity of Facebook news diets by measuring the proportion of news posts from sources across the political spectrum. We assign quality (factual, misinformation, or mixed) and political bias (left, center, or right) labels at the source level for each news domain. All posts originating from a

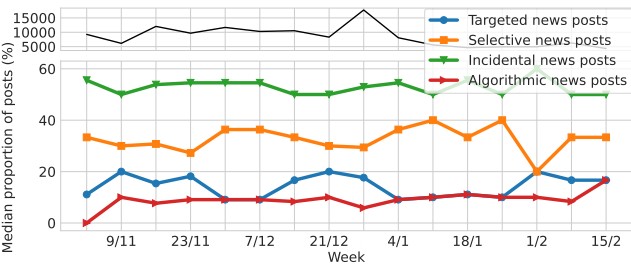

**Figure 8: Median proportion of selective, incidental, algorithmic, and targeted news posts received by users by week. Above: weekly number of collected news posts.**

specific domain inherit these labels. We do not make any judgment on the quality and political bias of news domains. Instead, similar to previous work [10], we rely on evaluations provided by Media Bias Fact Check and News Guard.

***Metadata on misinformation.*** News Guard describes whether a news source has a history of sharing misinformation in the "Topics" column of their data file, while Media Bias Fact Check provides it in the "Detailed" section of their website's source evaluation. Though the two agencies used different terminology to capture the full spectrum of misleading news practices, they always included the terms "Conspiracy," "Fake News," or "Misinformation." Consequently, we flagged a source as spreading misinformation if one of these terms was used to describe it. The two data providers agreed on this measure; only 33 domains were the subject of disagreement. We resolved these disagreements by applying the misinformation label. Overall, we labeled 456 news sources associated with 467 as spreading misinformation.

***Metadata on factualness.*** News Guard assigns a credibility score (between 0 and 100) and Media Bias Fact Check provides a factual_reporting text label for each news publisher. We apply filters to both fields: (a) News Guard scores of 75 or higher (indicating that a news source has high credibility or is generally credible), and (b) a positive Media Bias Fact Check factual reporting (High, Very high, or Mostly factual). If a news publisher has ratings from both agencies, we consider it factual only if both consider it factual. If a news publisher has an evaluation from only one agency, we use that rating alone. Overall, we have labeled 2,942 news sources as factual, corresponding to 3,074 Facebook pages.

***Metadata on news sources political bias.*** News Guard provides the political leaning for 2,939 different news sites (Far Left, Slightly Left, Center, Slightly Right, Far Right) and Media Bias Fact Check for 1,711 different news sites (extreme-left, far-left, left, left-center, center, right-center, right, far-right, extreme-right). We normalize the evaluations from both sources by keeping the News Guard scale, and we cast the Media Bias Fact Check evaluations into it by considering (a) extreme-right and extreme-left as far-right and far-left, (b) right as far-right and left as far-left and (c) right-center as slightly-right and left-center as slightly-left. We have 41 domains for which we have different evaluations from the two sources; we prefer to use the values of Media Bias Fact Check for these.

In total, from the 4,149 news sites in the Established News Sites list, we have the political leaning for 4,107 of them (99%). 64% of them are rated as Center, 20% as Left (7% Far-Left + 13% Slightly-Left), and 17% as Right (10% Far-Right + 7% Slightly-Right).

