# OpenReview forum: "What News Do People Get on Social Media? Analyzing Exposure and Consumption of News through Data Donations"
_ACM.org/TheWebConf/2024/Conference — TheWebConf24 Oral_

### Official Review · Reviewer_mDu1 · 2023-11-21

**Novelty:** 6
**Technical Quality:** 6

**Review:**

The paper addresses the problem of limited data access due to closed social media platforms in studying the impact of social media exposure on public discourse and political polarization. As a solution, the authors propose a privacy-preserving tool that captures users' interactions with news on Facebook to facilitate the donation of personal data for scientific study. The work recruits and collects data from 472 U.S. participants. Then, the authors investigate user interactions across four types of news exposure (selective, incidental, algorithmic, and targeted), two types of interaction (visible and hidden), and three types of news sources (factual, fake, and under-the-dog). The study reveals insights such as users encountering misinformation due to their active selection of low-quality news sources rather than exposure through friends or platform algorithms. Additionally, the study highlights users' willingness to engage with news sources holding opposing political ideologies, particularly in hidden interactions.

Strengths:
- S1. The authors put a great effort into developing a tool for auditing news exposure on Facebook. This tool proves crucial for overcoming the challenge of limited data access on closed social media platforms, contributing significantly to the research field.
- S2. The paper's methodology is robust. It constructs an extensive list of news sources and introduces the four types of news exposure and two types of interactions. These novel classifications enhance the depth and clarity of the study, providing a comprehensive understanding of users' interactions with news on social media.
- S3. The paper presents numerous interesting findings, which I note above.

I cannot think of any major reason to reject this paper, but I have minor concerns.

Weaknesses:
- W1. Limited News Domains: The inclusion of 404 news domains (with 449 Facebook pages) extracted from their own data might introduce bias, potentially leading to a higher fraction of posts from under-the-dog news sources.
- W2. The study indicates a liberal dominance among the participants, which could impact the generalizability of the findings. A more balanced participant representation across political ideologies would enhance the study's validity and applicability to a wider audience.

**Questions:**

NA

**Reviewer Confidence:**

3: The reviewer is confident but not certain that the evaluation is correct

**Scope:**

4: The work is relevant to the Web and to the track, and is of broad interest to the community

---

### Official Review · Reviewer_K7ga · 2023-11-22

**Novelty:** 6
**Technical Quality:** 6

**Review:**

This study proposes a tool to monitor user interaction with news on Facebook. Exploring this tool, the authors gathered user data and studied their behavior in this regard.

Positive aspects of the study:
- In general, the study is well-written and organized.
- The theme is relevant and useful for society.
- Interesting findings obtained.

Negative aspects of the study
- Some methodological steps are not clear – see comments/questions below.
- Small set of users. I understand it is hard to obtain the data explored in the study; however, the number of users could be bigger.
- Lack of details on the statistical tests performed. Also, lack of formality in different parts of the text.


Minor:
Inform a reference for “Prolific”.
Standard deviation is informative in conjunction with the mean instead of the median.

Perhaps Figure 2 would be easier to read as a CDF instead of a boxplot.

Indicate more clearly the visibility time part on fig2

There is a typo on the legend of fig 3.

Fig 8: Black line should be at the legend (it appears only on the caption); besides, the legend is covering it – it is hard to see part of it.

**Questions:**

Are the differences regarding visible and hidden interactions (2.6% and 2.8%) statistically different?

Regarding the time of exposure on users’ screens, did you treat cases where the users leave the page open for a long time, indicating they are not paying attention?

“ We further see that users performed both hidden and visible interactions on < 0.5% of posts, suggesting that the visible and hidden interactions are performed mostly on different sets of posts.” This was not clear.

**Ethics Review Description:**

-

**Reviewer Confidence:**

4: The reviewer is certain that the evaluation is correct and very familiar with the relevant literature

**Scope:**

4: The work is relevant to the Web and to the track, and is of broad interest to the community

---

### Official Review · Reviewer_v3QE · 2023-11-25

**Novelty:** 5
**Technical Quality:** 5

**Review:**

This study looks at the problem of news consumptoon on facebook, and answers how users find such content.

others approach in dealing with lack of data on both holsitic view of exposure and consumption is "data donation"
"methodology that can provide a
comprehensive picture of users’ news exposure on social media and
capture user interactions at a finer granularity"

which is through a tool called as CheckMyNews

472 users volunteered to share their data

provided source-level annotation to 8,084 more news sources

Author look at
- within which exposure type and what type of content can be found.
- within which exposure to how diverse content is
- how interaction type varies across content type, exposure type

The paper is well-written, easy to follow, and well-implemented.

**Questions:**

There are previous studies which introduce the ideas on various types of exposure, and self intrested of users. I suggest authors consider valusing their contribution, and as a support of literature on "a comprehensive picture of the news landscape" or "how users find content"


Robertson, Ronald E., Jon Green, Damian Ruck, Katherine Ognyanova, Christo Wilson, and David Lazer. n.d. “Engagement Outweighs Exposure to Partisan and Unreliable News within Google Search,” 23. Nature.  <== capture exposure


Subscriptions and external links help drive resentful users to alternative and extremist YouTube channels
AY Chen, B Nyhan, J Reifler, RE Robertson… - Science …, 2023 - science.org <== capture exposure

Examining the consumption of radical content on YouTube
Homa Hosseinmardi, Amir Ghasemian, Aaron Clauset, Markus Mobius, David M Rothschild, Duncan J Watts Proceedings of the National Academy of Sciences <== different referal modes, more than just algorithm, and introducing user preference

the finding are presented in break down of exposure type and interaction type.

granularity of data, surveys with metadata, all impresive.


In section 2.5, the authors mention they don't guarantee representativeness. My only concern is the heterogeneity of the findings across groups. I think they can simply check stratify on important variables such as partisanship and age, and replicate tables 1 and 2 with numbers. It can be in the Appendix.


My bet would be on differences in user engagement on mobile or desktop

"There
is currently no data suggesting that Facebook news diets on mobile
phones significantly differ from news diets on web browsers in
terms of composition, quality, and diversity of news posts."

But authors don't need to justify for that (convincing readers that they are perhaps the same). Just acknowledging it is a desktop. Researchers in this field are well aware of the difficulties and contributions for the desktop itself.


So everything under the list "Under-the-Radar Facebook Pages list" is potentially problematic?

**Reviewer Confidence:**

4: The reviewer is certain that the evaluation is correct and very familiar with the relevant literature

**Scope:**

4: The work is relevant to the Web and to the track, and is of broad interest to the community

---

### Official Review · Reviewer_DaQz · 2023-11-28

**Novelty:** 6
**Technical Quality:** 6

**Review:**

In this paper, the authors present a tool to collect data from users browsing Facebook. Users donate their traces to researchers, who analyze the data. This is similar to previous efforts, such as Facebook Tracking Exposed.  However, such data has not been analyzed before.

Pros

1. Very interesting work with real data about how users navigate and consume information on Facebook

2. Interesting findings on the distinction between consumption of news in an active versus reactive manner

3. Some discussion on the diversity of posts to which users are exposed to

Cons

1. There are previous works on tracking Facebook users, including Facebook Tracking Exposed, that the authors should refer to (see details below)

2. The authors set many conjectures that are hard to analyze because the data comes from real users with very different profiles (if everything changes from one user to another, what kind of general conclusions can we take?)

3. There is no baseline against which the authors compare their findings

4. The data may not be representative of real users -- less than 1,000 users are analyzed

I really enjoyed this paper.  My major concern relates to section 3.3 on the diversity of social media news. There are so many variables involved in the users profiles that it is very difficult to assess the root cause behind the level of diversity found in the users browsing behavior.  The approach taken in the following work, also using real data, seems more appropriate to answer the questions raised in section 3.3 about diversity:

Hargreaves, E., Agosti, C., Menasché, D., Neglia, G., Reiffers-Masson, A., & Altman, E. (2019). Fairness in online social network timelines: Measurements, models and mechanism design. Performance Evaluation, 129, 15-39.

Hargreaves, E., Mangabeira, E. F., Oliveira, J., Franca, T. C., & Mcnasché, D. S. (2020, December). Facebook News Feed personalization filter: a case study during the Brazilian elections. In 2020 IEEE/ACM International Conference on Advances in Social Networks Analysis and Mining (ASONAM) (pp. 615-618). IEEE.

In the above papers, authors use Facebook Tracking Exposed, by Claudio Agosti, to analyze the exposure of users to Facebook items: please, check the list of publications at https://facebook.tracking.exposed/

The idea of using neutral bots seems to me more appropriate to understand biases than the idea of using real users.

In Appendix C.2 the authors mention that they did not find evidence of bot behavior. However, leveraging bots may actually be necessary to validate some of the author's conjectures, as discussed above (in particular, on issues related to diversity, e.g., on Section 3.3).

**Questions:**

1) Are the authors familiar with Facebook Tracking Exposed and related works?

2) Did the authors consider using bots to assess the validity of some of their conjectures?

3) With respect to diversity of posts, could the authors discuss some of the fairness criteria behind Facebook choices? Please, check:

Hargreaves, E., Agosti, C., Menasché, D., Neglia, G., Reiffers-Masson, A., & Altman, E. (2019). Fairness in online social network timelines: Measurements, models and mechanism design. Performance Evaluation, 129, 15-39.

After the rebuttal phase, the reviewer best understood the similarities and distinctions between the present submission and the above work.  In particular, Hargreaves et al considered only time diversity (variations across timelines of a given user) to compute metrics of interest, whereas the current study considers space diversity (variations across users and relationships between users).

**Ethics Review Description:**

no ethics issues

**Reviewer Confidence:**

3: The reviewer is confident but not certain that the evaluation is correct

**Scope:**

4: The work is relevant to the Web and to the track, and is of broad interest to the community

---

### Decision · Program_Chairs · 2024-01-22

**Decision:**

Accept (Oral)

**Comment:**

This paper was reviewed by four members of the PC. Overall they gave scores of 6/6, 5/5, 6/6, 6/6 with reasonable levels of confidence.

 Reading through the comments, I find that there is reasonable excitement for this paper. The major comments about the number of users, I think, is fair, but an N<1000 is not unreasonable for a real-world survey.

 My biggest concern is the representativeness of the survey participants. Nevertheless, there is little that a researcher not at Facebook (or Twitter or Reddit) can do to avoid this.

 Speaking of Reddit. There is the "Consumers and Curators" paper by Glenski from a few years ago that likewise gets data donated from Reddit users. Perhaps its worth a brief note (or maybe some light comparison) in the final copy.